# Effects of Sugar Beet Silage, High-Moisture Corn, and Corn Silage Feed Supplementation on the Performance of Dairy Cows with Restricted Daily Access to Pasture

**DOI:** 10.3390/ani12192672

**Published:** 2022-10-05

**Authors:** José A. Aleixo, José Daza, Juan P. Keim, Ismael Castillo, Rubén G. Pulido

**Affiliations:** 1Graduate School, Faculty of Veterinary Sciences, Universidad Austral de Chile, P.O. Box 567, Valdivia, Chile; 2Animal Science Institute, Faculty of Veterinary Sciences, Universidad Austral de Chile, P.O. Box 567, Valdivia, Chile; 3Institute for Agricultural Research, Tamel Aike Research Centre, P.O. Box 296, Coyhaique, Chile; 4Animal Production Institute, Faculty of Agricultural Sciences, Universidad Austral de Chile, P.O. Box 567, Valdivia, Chile; 5Empresas Iansa, Osorno, Chile

**Keywords:** sugar beet silage, milk production, rumen fermentation, grazing

## Abstract

**Simple Summary:**

Seasonal variations in herbage growth rate and nutrient composition result in low herbage intake which is insufficient to meet the nutritional requirements of cows and limits milk production. The main opportunities for increasing milk production per cow and per ha are a careful pasture management to ensure adequate herbage high-quality and strategic low-cost and high-energy feed supplementation. Supplementation strategies include different combinations of level, type, and processing of supplemental feeds. This study evaluates the effect of supplementation with sugar beet silage, corn silage, or high-moisture corn (HMC) on dairy performance, rumen, and plasma metabolites in dairy cows under conditions of restricted grazing in spring. We found that the supplementation with sugar beet silage allowed milk production, live weight, and fat concentration similar to corn silage and HMC, but with a lower concentration of milk protein than HMC. Results suggest that sugar beet silage can be used as an alternative supplement for high-producing dairy cows under conditions of restricted grazing in the current experiment.

**Abstract:**

A study was undertaken to assess the effect of supplementation with sugar beet silage, corn silage, or high-moisture corn on dairy performance, rumen, and plasma metabolites in dairy cows under conditions of restricted grazing in spring. Eighteen multiparous Holstein Friesian cows, stratified for milk yield (39.4 kg/day ± 3.00), days of lactation (67.0 days ± 22.5), live weight (584 kg ± 38.0), and number of calves (5.0 ± 1.5), were allocated in a replicated 3 × 3 Latin square design. Treatments were as follows: SBS (10 kg DM of permanent pasture, 7 kg DM of sugar beet silage, 4 kg DM of concentrate, 0.3 kg DM of pasture silage, 0.21 kg of mineral supplement); corn silage (10 kg DM of permanent pasture, 7 kg DM of corn silage, 4 kg DM of concentrate, 0.3 kg DM of pasture silage, 0.21 kg of mineral supplement), and HMC (10 kg DM of permanent pasture, 5 kg DM of high-moisture corn, 4.5 kg DM of concentrate, 1.2 kg DM of pasture silage, 0.21 kg of mineral supplement). Pasture was offered rotationally from 9 a.m. to 4 p.m. Between afternoon and morning milking, the cows were housed receiving a partial mixed ration and water ad libitum. The effect of treatments on milk production, milk composition, body weight, rumen function, and blood parameters were analyzed using a linear–mixed model. Pasture dry matter intake (DMI) was lower in SBS than CS (*p* < 0.05) and similar to HMC, but total DMI was higher in HMC than SBS (*p* < 0.05) and similar to CS. Milk production for treatments (32.6, 31.7, and 33.4 kg/cow/day for SBS, CS, and HMC, respectively), live weight, and fat concentration were not modified by treatments, but milk protein concentration was lower for SBS compared with HMC (*p* < 0.05) and similar to CS. B-hydroxybutyrate, cholesterol, and albumin were not different among treatments (*p* > 0.05), while urea was higher in SBS, medium in CS silage, and lower in HMC (*p* < 0.001). Ruminal pH and the total VFA concentrations were not modified by treatments (*p* > 0.05), which averaged 6.45 and 102.03 mmol/L, respectively. However, an interaction was observed for total VFA concentration between treatment and sampling time (*p* < 0.05), showing that HMC produced more VFA at 10:00 p.m. compared with the other treatments. To conclude, the supplementation with sugar beet silage allowed a milk response and composition similar to corn silage and HMC, but with a lower concentration of milk protein than HMC. In addition, sugar beet silage can be used as an alternative supplement for high-producing dairy cows with restricted access to grazing during spring.

## 1. Introduction

Grazing pasture is the basis for dairy production systems in regions with temperate climates, such as Ireland, New Zealand, parts of Australia, United States, Europe, and South America, due to its low cost compared with other feed ingredients for dairy cows [1]. However, herbage variations on growth rate and nutrient composition occur, and limited total DMI makes herbage insufficient for animal requirements, especially in dairy cows in early lactation or with high yielding levels [2,3]. Restricted pasture access can be used as a management tool to increase grazing efficiency and pasture utilization [4,5]. However, pasture restriction limits pasture DM intake (DMI) and milk production by dairy cows [6]. Therefore, it is necessary to supplement feeds to meet the nutritional requirements of dairy cows [7].

A recent review stated the main opportunities to increase milk production per cow and per ha are as follows: (a) careful pasture management to ensure adequate herbage high quality and (b) strategic supplementation with low-cost and high energy feeds [8]. The energy supplementation strategies include a combination of level, type, and processing feeds [9]. In addition, when supplements are offered as a partial mixed ration (PMR), the improvement in milk response is due to a less variable pH in ruminal fluid, more stable and efficient rumen fermentation, and increased pasture DMI in cows with PMR compared with feeding the same amount of dietary energy as grain in the milking parlor or through forage in feeders [10].

There are several sources of energy supplementation; corn is the main energy source in diets of high-producing dairy cows because it is a cost-effective source of digestible energy to increase milk production [2]. Productive responses of lactating dairy cows to corn-based supplements depend on the dietary inclusion level, the basal ration, the physical processing, and the potential genetic of the dairy cow [11].

Sugar beets (*Beta vulgaris* ssp. vulgaris) are a well-established crop in many parts of the world (North and South America, Europe, Oceania/Australia), and are characterized by high yields per hectare. Although sugar beets and their byproducts are primarily grown for sugar production, they have been used as feed for cattle for centuries. Beets have a very high sugar concentrations, contain little ash, and have high DM and NDF digestibility; therefore, they have a high net energy content compared with other energy feed sources [12,13,14]. Traditionally, beets have been used as feed during winter, but to be available for feed all year round, they require conservation, e.g., ensiling [13].

Replacing energy grain and forages with sugar beet in dairy cow diets has been studied over recent decades, with contradictory results [12,15,16]. Furthermore, in the review performed by Evans and Messerschmit [14], they stated that there are few feeding trials to support sugar beets as a partial replacement for energy sources in rations for high-producing dairy cows. Thus, the aim of the present research was to evaluate the effect of supplementation with sugar beet silage, corn silage, or high-moisture corn on milk production and composition, ruminal fermentation, and blood indicators for energy and protein metabolism of dairy cows given restricted daily access to pasture.

## 2. Materials and Methods

The study was carried out between 21 October and 23 December 2019 at Austral Agricultural Research Station of the Universidad Austral de Chile (39°47′ S, 73°14′ W). The animal-handling procedures described in this study were approved by the Animal Welfare Committee of the Universidad Austral de Chile (grant number C26-2020).

### 2.1. Animals and Study Design

Eighteen multiparous lactating Holstein Friesian cows were used in a replicated 3 × 3 Latin square design, with three treatments and three time periods. Each period had a 21-day duration, where the first 14 days of each period were for adaptation to the diets, and the last 7 days were for data and sample collection. Cows were grouped according to milk production (39.4 ± 3.0 kg/d), body weight (BW; 584 ± 38.0 kg), and days in milk (DIM; 67.0 ± 22.45). They were randomly allocated to one of three treatments: (1) SBS—10 kg DM of pasture, 6.5 kg DM sugar beet silage, 0.3 kg DM of grass silage, 3.5 kg concentrate, and 0.21 kg of mineral salts; (2) CS—10 kg DM of pasture, 6.5 kg DM corn silage, 0.3 kg DM of grass silage, 3.5 kg concentrate, and 0.21 kg of mineral salts; (3) HMC—10 kg DM of pasture, 5.0 kg DM high moister corn, 1.8 kg DM of grass silage, 4.0 kg concentrate, and 0.21 kg of mineral salts. Diets were formulated to be isoenergetic and isonitrogenous. A summary of the composition of diets at the beginning of the study is presented in Table 1.

### 2.2. Grazing Management

Treatments were strip-grazed in the same paddock, separated by an electric fence. Herbage allowance were offered to cows (measured at ground level) of 18 kg DM cow/d, which was allocated daily after morning milking (08:00 h), allowing them to graze for 7 h. The daily area to be grazed form each treatment was adjusted by herbage allowance and pre-grazing herbage mass (HM). Pre-grazing HM (kg DM/ha, above ground level) was estimated using a rising plate meter (RPM; Ashgrove Plate Meter, Hamilton, New Zealand). A total of 100 RPM measurements were taken in each grazing area allocated to each treatment; then, HM was estimated using a specific equation for spring pastures in Southern Chile (HM kg DM/ha = RPM × 100 + 400; R^2^ = 0.75) [17]. Post-grazing HM was estimated using the same methodology.

### 2.3. Pasture and Supplements

The study was carried out on 10 ha of permanent pasture dominated by L. perenne, sown under irrigation. Sward was divided in paddocks which were 50 m wide, which were subdivided in strips located between 500 and 1000 m from the milking parlor. Grass silage was produced from pastures harvested at the Austral Agricultural Research Station. Briefly, pasture was cut down and then withered for 24 h before being stored and preserved in plastic bales. Sugar beet silage was a type of bolus, which was made with beet roots, chopped, and bagged. At the time of chopping, the additive (Ecosyl^TM^) and an absorbent additive (alfalfa hay, in proportion to 20% of the DM) were added to the silage. The additive (Ecosyl^TM^) for silage contains Lactibacillus plantarum (MTD/1) NCIMB40027: −min 2.50 × 1011 cfu/g The recommended dose for the product provides a total of 500,000 cfu/g of ensiled green matter. Maize silage particles with an approximate size of 1.0 cm were stored and preserved in silage stacks; then, they were rolled and sealed on the same day as they were harvested. Concentrate was produced by IANSAGRO S.A., which was individually delivered to the animals during the morning and afternoon milking (0.5 and 6.0 kg DM/animal, respectively).

All cows were offered the silages (sugar beet, corn silage, HMC, and pasture silage), the concentrate, and the mineral salts as a PMR in feeding pens after p.m. milking in the cowshed. Water was consistently freely accessible in the paddock and in the cowshed.

Pasture samples were collected in triplicates in each study period; each collection was performed before the cows entered the new daily strip, using the hand-plucking technique in locations where pasture was cut at 4 cm height. The content of DM in the feeds was determined on days 1, 3, and 5 of week 3 of each period. All samples were frozen at −20 °C until the laboratory analysis. Samples of pasture, concentrate, and silages were freeze-dried and ground through a 1 mm sieve (Willey Mill, 158 Arthur H, Thomas, Philadelphia, PA, USA). For each sample, ash and lipids were analyzed according to [18] (methods 942.05 and 920.39 for ash and lipids, respectively); nitrogen (N) content was determined by combustion (Leco Model FP-428 Nitrogen Determinator; Leco Corporation, St Joseph, MI, USA) and was used to calculate CP content (N × 6.25). Neutral detergent fiber (NDF) was determined [19] using heat stable amylase (Ankom Technology Corp., Macedon, NY, USA) and ADF was determined according to [18] (method 973.18). Metabolizable energy was estimated using the regression by “D” value (digestible organic matter/DM 100) and determined in vitro [20] according to Goering and Van Soest [21].

### 2.4. Milk Production and Body Weight

Cows were milked at 07:00 h and 16:00 h and milk yield was recorded daily with a flow sensor (MPC580 DeLaval, Tumba, Sweden) during the study periods. The average for the final week of each period is reported here. Representative milk samples were collected at morning and afternoon milking for 3 days in the last week of the study period to assess fat, protein, and milk urea contents, as analyzed by mid-IR spectrophotometry (Foss 4300 Milko-scan; Foss Electric, Hillerod, Denmark). Body weight was recorded daily after both milking sessions with an automated weighing scale (AWS100 DeLaval, Tumba, Sweden) at the exit of the milking parlor.

### 2.5. Dry Matter Intake

The silages and concentrates offered and refusals were recorded daily during the last week of each grazing period. A subsample was taken to determine their DM content; thus, the silages and concentrate DMI were determined.

To estimate pasture DMI, cows were divided in pairs (according to their respective experimental square) and a single strip was offered to each pair of cows. Apparent herbage DMI was estimated according to Bryant et al. [22], as follows:DMI = [Pre mass (kg DM/ha) − post mass (kg DM/ha)] × grazing area (ha)/No. of animals

Finally, total DMI was calculated as the summation of pasture DMI, silages, and concentrates.

### 2.6. Blood and Rumen Parameters

Blood samples were collected at 16:30 h (afternoon milking) and 21:30 h on day 4 in the last week of study period. Samples were collected using vacutainers with no anticoagulant from coccygeal vessels, which were immediately transported on ice to the hematology laboratory of the Universidad Austral de Chile. In the laboratory, blood samples were centrifuged at 800 × g for 10 min to obtain the plasma, which was frozen at –20 °C for subsequent analyses. Plasma was used to measure the concentration of b-hydroxybutyrate (Ranbut; Randox Laboratories, Crumlin, County Antrim, Northern Ireland), urea (glutamate dehydrogenase, HUMAN, Wiesbaden, Germany), and albumin (BGC method) by using a commercial kit (Human Albumin Liquicolor) using a Metrolab 2300^®^ auto analyzer (Wiener Laboratory, Rosario, Argentina).

Samples of rumen fluid (20 mL) were collected using a stomach tube (Flora Rumen Scoop; Prof-Products, Guelph, ON, Canada) at 15:30 h and 21:30 h on day 20 in each experimental period. After collection, samples were strained through 4 layers of cheesecloth and 10 mL of rumen fluid was drawn off, mixed with 0.2 mL 50% (*w*/*v*) sulfuric acid, and stored at −20 °C until laboratory analysis. Samples were thawed for 16 h to 4 °C, and then centrifuged at 10,000× g for 10 min at 4 °C. Six milliliters of supernatant was drawn off and centrifuged at 10,000× g for 10 min at 4 °C. Thawed supernatant of rumen fluid samples were analyzed for VFA by gas chromatography (GC; Shimadzu GC-2010 Plus High-end gas chromatography, equipped with gas chromatography capillary column, SGE, BP21 (FFAP); Shimadzu Corporation, Kyoto, Japan), as described by Tavendale et al. [23].

### 2.7. Statistical Analysis

The effects of treatments on milk production, milk composition, body weight, rumen function, and blood parameters were analyzed using a linear–mixed model (LMM) in R software (R Foundation for Statistical Computing, Vienna, Austria). The model included the fixed effects of treatment and period, and the random effect of cows nested within the squares. For rumen fatty acids, the hour of sampling was included as a repeated measurement and the interaction of treatment with the repeated measurement was accounted for. The level of significance was declared at *p* < 0.05, and a significant multiple comparison test was performed.

## 3. Results

### 3.1. Chemical Composition of Food

The chemical compositions of the pasture samples and the supplements are presented in Table 2. The chemical composition of the pasture samples was similar among paddocks (*p* > 0.05). Grass silage, corn silage, and HMC have similar averages of DM, CP, and ME. Metabolizable energy content of sugar beet silage was similar to that of corn silage, but it had greater CP content compared with corn silage and HMC.

### 3.2. Grazing Management and DMI

The results of the grazing management and DMI are presented in Table 3. Pre-grazing herbage mass was similar among treatments (*p* > 0.05), but post-grazing herbage mass was greater for SBS and lower for HMC. Pasture DMI was greater for CS-fed cows and lower SBS-fed cows (*p* < 0.05). Silage DMIs were greater for SBS-fed cows, intermediate for CS-fed cows, and lower for HMC-fed cows (*p* < 0.05). Consequently, total DMI was high in cows supplemented with HMC compared with those supplemented with SBS but was similar to those supplemented with CS (*p* > 0.05). The CP intake was greater in SBS-fed cows compared with HMC-fed cows (*p* < 0.05), but was similar to CS-fed cows. The intake of ME was greater for CS- and HMC-fed cows compared with SBS-fed cows (*p* < 0.05). Intake of NDF was high for CS-fed cows, lower for HMC-fed cows, and intermediate for SBS-fed cows (*p* < 0.001).

### 3.3. Milk Production and Body Weight

The results of milk production and body weight are presented in Table 4. Milk production did not differ among treatments, averaging at 32.7 kg/d. Similarly, fat in milk (as % and g/d) and milk urea nitrogen were not modified by dietary treatments (*p* > 0.05), whereas the content of protein in milk was greater (*p* < 0.05); the production of protein in milk tended to be greater in HMC-fed cows (*p* = 0.052). Body weight did not differ among treatments (*p* > 0.05).

### 3.4. Blood and Ruminal Parameters

The results of blood parameters are presented in Table 5. Plasma concentration of BHB tended to be lower in HMC-fed cows compared with SBS-fed cows (*p* = 0.077). Cholesterol and albumin did not differ among the treatments (*p* > 0.05). Plasma concentration of urea nitrogen was high in SBS-fed cows, intermediate in CS-fed cows, and lower in HMC-fed cows (*p* < 0.01).

Results of rumen fermentation parameters are presented in Table 6. Ruminal pH and the production and the concentrations of all VFAs did not differ among treatments. However, an effect of the time of day (hour of sampling) on total VFA was observed; additionally, we observed an effect on the ruminal concentration of propionate, butyrate, and valerate (*p* < 0.01), which were 24, 54, and 38% greater at night-time sampling compared with the afternoon sampling. We observed an interaction between treatment and time of day for total rumen VFA concentration (*p* < 0.01), with a greater concentration of total AGV in the HMC-supplemented cows.

## 4. Discussion

To our knowledge, this is the first study evaluating the effect of supplementation with sugar beet silage, corn silage, or high-moisture corn offered on a PMR on animal performance and metabolism of high-yielding cows grazing for a restricted time during the day. In addition, research in supplementation with sugar beet silage is limited, with research being focused on cows fed with TMR [13,24].

It is known that the tools for controlling grazing among dairy cows include the use of grazing frequency and herbage allowances defined for each grazing season of the year. In addition, the daily supply of pasture and pre-grazing herbage mass affect pasture DMI [7]. For this study, a pasture allowance at ground level of 18 kg DM cow/day was offered to achieve a pasture consumption of 10 kg DM cow/day. However, pasture DMI was slightly lower for SBS and greater for HMC; these results agree with the greater post-grazing herbage mass obtained in SBS and HMC treatments, which directly affected the total DMI and the intake of ME in those treatments. It is reported that the time spent chewing during eating and ruminating increased with fiber and particle size [14]. Then, lower DMI of pasture in the SBS treatment could be explained by the greater volume in the rumen of pieces of sugar beet silage used to ruminate in this group of cows; this gave a feeling of satiety in the cows because the rumen was filled; consequently, the cows’ desire to graze was reduced. Keim et al. [25] observed a reduction in DMI with the inclusion of sugar beet in the diet, which was associated with a greater eating time due to the physical structure (difficult to eat) of sugar beet roots. Furthermore, it is known that there is variability with respect to the rate and degree of degradation and end products of carbohydrate fermentation [26]. Therefore, the slower fermentation of the cell walls of the sugar beet silage and the generation of VFA during the rumination process could have generated a degree of satiety with an effect on the pasture DMI. Finally, there was a consistent effect of the ingredients of the PMR on the pasture intake [10].

### 4.1. Milk Production and Composition

The achievement of high milk yields among dairy cows depends on high DM intake and energy concentration in the rations. To achieve the nutritional requirements of high-yielding dairy cows, energy-rich diets are required. Forages are often replaced by grains such as corn to increase feed and energy intake. In this study, milk production did not differ among treatments, averaging 32.7 kg/d. In the work of Hellwing et al. [13], the replacement of sugar beet silages differing in fermentation intensity with a conventional maize silage showed a decrease in milk yield. Later, El Tawab et al. [24] studied the effect of partial or complete replacement of corn silage with ensiled sugar beet tops, and observed no effect on daily milk production or feed efficiency. Literature reports differences in milk production with sugar beet silage feed; this is possibly due to differences in the silage composition (pure beet silages or co-ensiled with other ingredients) and to differences in feed ingredients substituted with sugar beet silage. In addition, it is probable that, in this study, the similar milk production could be explained by contents of ME and CP; while these levels were different to those used in the literature, the difference might not have been enough to produce changes in the milk yields of the cows. Finally, we can also speculate that a mobilization of body reserves by the animals to support the lactogenesis demands might have influenced our results. Unfortunately, body condition change was not evaluated in this study due to the characteristics of our design that did not allow us to validate this hypothesis.

Feeding the cows with SBS did not modify the fat (as % and g/d) or milk urea nitrogen in the yielded milk compared with the other treatments. In general, sugar beets can be viewed as a mixture of beet pulp and sucrose [14]. An increased sucrose supply through supplementation with sugar beet has previously been shown to increase milk fat concentration [13,27,28]. Recently, Keim et al. [25] observed an increase in milk fat content when sugar beet roots replaced corn in the diets of pasture-fed dairy cows. Schmidt et al. [15] found a lower concentration of milk fat when molasses and barley were replaced with sugar beet silage. Kirchgessner et al. [28] concluded that the net energy content of sucrose was lower than expected from the content of digestible energy.

The content of protein in milk in this study was lower with SBS supplementation; this result disagrees with those of Hellwing et al. [13]. Although the amount of ingested CP was higher in SBS-supplemented cows, this was not reflected in a higher amount of urea in the milk, nor in a higher concentration of milk protein, due to a lower ME intake. It has been pointed out that, for HMC supplementation, there would be a greater contribution of energy to the rumen microorganisms because of the greater ruminal degradability and the total digestibility of starch [29], and due to an increase in microbial access to starch granules. Therefore, it is expected that, at the ruminal level, the energy concentration would have better synchrony with the CP from the pasture, decreasing the urea concentration and increasing protein concentration in milk; this hypothesis could not be corroborated in this study because ruminal NH_3_ was not measured.

### 4.2. Blood and Ruminal Parameters

Blood parameters were used to assess livestock health, nutritional, physiological, and pathological status. Metabolic disorders of nutritional origin are frequent in the first months of lactation and during the spring in high-producing grazing dairy cows [30]. In the present experiment, plasma ketones, measured as concentrations of BHB, tended to be different between treatments and averaged 0.9 mmol/L (Table 6); this is over the recommended reference range (0.1–0.6 mmol/L) for dairy cows in early lactation [30] and considered as a trend of negative energy balance (between ≥0.6 and <1.2 mmol/L) [31]. The greater concentration of plasma BHB in cows receiving sugar beet silage could be explained by both the lower intake of DM and ME and the intake of the sugar beet silage itself being richer in sucrose [14].

It has been reported [32,33] that greater ruminal synchrony between ruminal degradable protein and energy allows greater utilization of dietary protein by ruminal microorganisms. Plasma concentrations of urea were different among treatments, being greater in SBS-fed cows, intermediate in CS-fed cows and lower in HMC-fed cows (*p* < 0.01); however, the reported concentrations were within the reference ranges reported by Wittwer [30]. These results suggest that NH_3_-N utilization by rumen microbes was more efficient in HMC-supplemented cows, due to its lower CP and greater ME intake. The slower values of degradability exhibited by HMC compared with corn silage—because of corn structure and density—would have synchronized better over time with high-degradable protein by consumed these pastures through grazing [32]. The cholesterol and albumin concentrations were similar among treatments and within the reference range, indicating an adequate nutritive balance in the treatments for those variables [33].

Unlike grains, where the primary carbohydrate is starch, the majority component of carbohydrates in beets is sugar. With sugar feeding, the primary concern is the perception that it will ferment to acids quickly, decreasing rumen pH and contributing to acidosis [12]. Ruminal pH and the production and concentrations of all VFAs did not differ among treatments, and the ruminal pH of all treatments fell to within the acceptable levels for optimum fiber digestion.

In grazing dairy cows, rumen fluid concentrations of VFAs can range between 50 and 150 mmol/L, depending on the diet [34]. In our study, the average total VFA production was 102.0 mmol/L, which is in the over-middle part of the range reported by Holmes et al. [34]; this is probably because of the good synchrony between the CP and the energy of the diet. Nevertheless, an effect of time of day (end of grazing (4:00 p.m.) and the end of intake of PMR (10:00 p.m.)) on evening sampling was observed, with greater concentrations of total VFA; the ruminal concentrations of propionate, butyrate, and valerate could be explained by both the higher intake of starch and sugars from the silages and the better synchrony between CP and energy from eating the PMR compared with the intake of pasture.

Finally, an interaction between treatment and sampling time was observed, with a greater concentration of total AGV in the HMC-supplemented cows; this can be seen as a product of the consumption of the TMR, which is rich in carbohydrates. The carbohydrates content lends TMR a greater digestibility than the other treatments; thus, it is associated with a greater energy contribution to the rumen of microorganisms being made by the high-moisture corn.

## 5. Conclusions

Supplementation with sugar beet silage demonstrated a response in milk yield, showing a composition similar to those of corn silage and HMC supplementations, but with a lower concentration of milk protein than HMC supplementation. In addition, based on the results of this study, sugar beet silage can be used as an alternative supplementation for highly productive cows with restricted access to grazing during spring.

## Figures and Tables

**Table 1 animals-12-02672-t001:** Ingredients and chemical composition of treatments.

	Treatments ^2^		
	SBS	CS	HMC
Ingredient ^1^			
CP, % DM	15.2	15.5	15.3
NDF, % DM	33.7	35.9	29.5
ADF, % DM	16.8	16.4	13.1
Lipid, % DM	1.7	2.3	2.2
ME, Mcal ME/kg DM	2.70	2.78	2.76
Ash, % DM	7.0	6.8	5.5

^1^ DM—dry matter; CP—crude protein; NDF—neutral detergent fiber; ADF—acid detergent fiber; ME—metabolizable energy. ^2^ SBS: sugar beet silage; CS: corn silage; HMC: high moister corn.

**Table 2 animals-12-02672-t002:** Chemical composition of feeds offered to grazing dairy cows during the study.

Item	Feeds					
	Silages				Pasture	Concentrate
	SBS	CS	HMC	Pasture Silage		
DM, %	28.9 ± 1.43	39.5 ± 5.06	72.7 ± 2.62	28.3 ± 1.43	25.1 ± 5.77	91.5 ± 1.26
Ash, %	6.7 ± 0.02	5.7 ± 0.86	1.2 ± 0.03	6.9 ± 0.32	7.75 ± 0.66	4.6 ± 0.15
CP, %	12.8 ± 0.36	7.2 ± 0.23	6.9 ± 0.10	14.1 ± 0.27	18.1 ± 2.86	21.7 ± 1.15
NDF, %	29.0 ± 0.88	36.7 ± 3.07	8.2 ± 0.24	44.0 ± 2.92	39.9 ± 1.23	20.0 ± 0.84
ADF, %	19.1 ± 0.46	20.0 ± 1.44	2.6 ± 0.19	30.1 ± 0.35	17.7 ± 0.74	7.1 ± 0.23
EE, %	0.7 ± 0.06	2.3 ± 0.22	2.9 ± 0.12	2.2 ± 0.42	1.56 ± 0.15	3.8 ± 0.11
ME, Mcal/kg DM	2.69 ± 0.01	2.65 ± 0.04	2.95 ± 0.01	2.40 ± 0.01	2.74 ± 0.06	2.80 ± 0.01
NFC, %	50.8 ± 0.58	48.1 ± 2.21	80.7 ± 0.26	32.8 ± 2.78	32.8 ± 3.87	49.1 ± 0.78
Starch, %	-	30.8 ± 3.42	71.2 ± 1.16	-	-	32.2 ± 0.61
pH	3.44 ± 0.11	3.4 ± 0.06	5.0 ± 0.08	3.6 ± 0.10	-	-
N-NH_3_, % total N	5.55 ± 0.23	8.9 ± 0.75	9.89 ± 1.89	8.0 ± 0.92	-	-
Lactic Acid, %	2.22	2.46	5.87	4.10	-	-
Acetic Acid, %	0.54	0.65	1.02	1.21	-	-

DM—dry matter (%); CP—crude protein (% DM); NDF—neutral detergent fiber (% DM); ADF—acid detergent fiber (% DM); ME—metabolizable energy (Mcal/kg DM); NFC—nonfibrous carbohydrates (% DM); NH_3_-N—ammonia-N (%), acetate, and butyrate (mmol/L).

**Table 3 animals-12-02672-t003:** Grazing management and dry mater intake (DMI) of grazing dairy cows supplemented with PMR composited by sugar beet silage, corn silage, or HMC.

	Treatment			SEM ^1^	*p*-Value
	SBS	CS	HMC		
Herbage mass, kg DM/ha					
Pre-grazing	3368	3415	3422	61.79	0.7609
Post-grazing	2074a	2002ab	1956b	46.05	0.0471
Intake, kg DM/cow/d					
Pasture	9.46b	10.31a	10.20ab	0.243	0.029
Treatment silage	6.27a	5.95b	5.03c	0.059	<0.001
Grass silage	0.29b	0.15c	1.42a	0.009	<0.001
Concentrate	3.53	3.83	3.83	-	-
Total	19.55b	20.19ab	20.54a	0.249	0.020
Nutrient intake					
CP, kg CP/d	3.36a	3.24ab	3.20b	0.047	0.039
ME, Mcal/d	52.96b	55.77a	56.73a	0.687	0.001
NDF, kg NDF/d	6.56b	7.05a	5.87c	0.098	<0.001

^1^—SEM: Standard error of the mean; DM: dry matter. a, b, c within a row, different letters re resent the significant differences at *p*-value < 0.05.

**Table 4 animals-12-02672-t004:** Milk production and body weight of grazing dairy cows supplemented with PMR composited by sugar beet silage, corn silage, or HMC.

	Treatments			SEM ^1^	*p*-Value
	SBS	CS	HMC		
Milk production, kg/d	32.68	31.77	33.43	0.686	0.297
Feed efficiency, milk yield/feed intake	1.67	1.57	1.62	-	-
Milk fat content, %	4.18	4.42	3.99	0.144	0.152
Milk protein content, %	2.97b	3.11ab	3.21a	0.045	0.006
Milk urea N, mg/L	279	257	237	13.80	0.152
Milk Fat, kg/d	1.36	1.39	1.33	0.052	0.750
Milk protein, kg/d	0.97	0.99	1.07	0.031	0.052
Body weight, kg	593	595	594	8.41	0.988

^1^—SEM: Standard error of the mean. a, b, c within a row, different letters re resent the significant differences at *p*-value < 0.05.

**Table 5 animals-12-02672-t005:** Blood parameters of grazing dairy cows supplemented with PMR composited by sugar beet silage, corn silage, or HMC.

	Treatment			SEM ^1^	*p*-Value
	SBS	CS	HMC		
BHB, mmol/L	1.01	0.92	0.79	0.062	0.077
Cholesterol, mmol/L	4.57	4.74	4.18	0.198	0.186
Blood urea N, mmol/L	4.89a	3.88b	2.94c	0.178	<0.001
Albumin, g/L	33.17	33.67	32.79	0.688	0.706

^1^—SEM: Standard error of the mean.

**Table 6 animals-12-02672-t006:** Rumen fermentation parameters of grazing dairy cows supplemented with PMR composited by sugar beet silage, corn silage, or HMC.

	Treatment			Time of Day		SEM ^1^	*p*-Value		
	SBS	CS	HMC	Am	Pm		Trt	Time	Int
pH	6.45	6.52	6.39	6.58	6.33	0.03	0.463	<0.01	0.46
VFA, mmol/L									
Acetate	40.46	39.47	45.64	39.03	44.69	1.51	0.363	0.06	0.234
Propionate	22.29	21.02	20.50	19.01	23.53	0.49	0.355	<0.01	0.090
Butyrate	21.26	20.10	20.50	16.29	24.95	0.76	0.827	<0.01	0.355
Valeric	9.61	5.81	6.39	6.13	8.38	1.06	0.302	0.298	0.305
Isobutyrate	6.80	6.07	5.55	4.76	7.52	0.46	0.586	<0.01	0.456
Isovaleric	5.85	5.03	3.69	3.02	6.69	0.59	0.419	<0.01	0.618
Total	106.29	97.52	102.28	88.26	115.79	2.23	0.306	<0.01	0.010

^1^—SEM: Standard error of the mean.

## Data Availability

Data presented in this study are available requesting the corresponding author.

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
