# Peer review of "Effects of Sugar Beet Silage, High-Moisture Corn, and Corn Silage Feed Supplementation on the Performance of Dairy Cows with Restricted Daily Access to Pasture"

_animals, 2022, doi:10.3390/ani12192672_

Round 1

Reviewer 1 Report

The experiment is quite novel and does have some novel points. However, there are some issues which have been highlighted below but also in the attached text. 

1. In text references is not in journal format and should be adjusted throughout.

2. Table 3 should be revised to have some distinction between the two pasture. 

Author Response

Dear reviewer,

We would like to thank for your suggestions, which improved the quality of manuscript. The original text has modifications and we have attended your comments.

Answer for each comment was included, as described below.

regards

Dr. Rubén Guillermo Pulido

Correspondence author

ANSWERS Reviewer

L21: change text “sugar beet silage allowed a milk production” by “sugar beet silage allowed milk production”

L30: change text “0.3 DM kg of pasture silage” by “0.3 kg DM of pasture”

L166 change text “Body weight was daily recorded after both milking with” by “Body weight was recorded daily after both milking with”

L219: Table 3. Change on feeds text “Pasture” by “Pasture Silage”

L219: Table 3.

Comment: These parameters can be placed in another table for rumen environment.

Answer: These parameters are at the silage fermentation.

L390-460: Text and References according to Animals style

Reviewer 2 Report

In the current study, the authors examined the effect of the inclusion of sugar beet silage, corn silage or high moisture corn, on performance, rumen, and plasma metabolites in dairy cows during periods (spring) of forage shortage.

The findings are very promising since the literature is limited regarding the use of sugar beet silage in dairy nutrition. In general, the paper has a notable structure, even though English editing is strongly recommended throughout the manuscript. The aim of the work is clear, and the results are well-presented. However, I found some crucial flaws that prevent a positive suggestion for paper’s publication in the present form.

1)     The English must be improved in the whole manuscript

2)     The number of animals used for this study is too low (18), making the experimental design very weak and thus, the findings cannot strongly be supported

3)     A final paragraph in the discussion section, highlighting the main findings of this work, as well as the benefits the industry and stakeholders may obtain by these findings, is recommended.

Comments

Introduction

Line 63-64: instead of “However, pasture restriction limits pasture DM intake (DMI) and milk production by dairy cows (Mattiauda et al., 2013), therefore, it is necessary to provide supplementary feeds to meet the nutritional requirements of the milking herd (Peyraud and Delagarde, 2011)”, it is better to write “However, pasture restriction limits pasture DM intake (DMI) and milk production in dairy cows (Mattiauda et al., 2013). Therefore, it is necessary to provide dairy animals with supplementary feeds that meet their nutritional requirements (Peyraud and Delagarde, 2011)”.

Line 67: delete “has”

Line 68: instead of “as” write “the”

Lines 72-76: Rephrase by making shorter sentences; Also Line 72: instead of “were” write “are”, Line 73 you mean “improvements” instead of” improves”.

Lines 77-79: Split the sentence in lines 77-79 in 2 sentences. Write “There are several sources of energy supplementation, with corn being the main energy source in diets of high-producing dairy cows.” rephrase the rest “its cost-effective source of digestible energy to increase milk production (Bargo et al., 2003)”

Line 85 - 86: instead of “Although primarily grown for sugar production, sugar beets and their by-products have been used as feed for cattle for centuries”, write “Although sugar beets and their by-products were primarily grown for sugar production, they used as feeds for cattle for centuries”

Line 90: “fed fresh” to whom? Please specify, instead of “but to be fed all year round require conservation”, write “but to be fed all year round, a conservation method is required”

Materials & Methods

Lines 111-118: provide the formulation of different feeds in Table 1 and delete it from the text, Table 1 should be “Ingredients and chemical composition of treatments” presented in %

Line 124: delete the “n” from “an”

Line 126: “group” instead of “treatment”

Line 142: “of” instead of “for”, “contained” instead of “contains”

Line 143: “was provided to be” instead of “provides a total of”

Line 149 -151: rephrase

Line 152-153: “in triplicates” instead of “3 times”, “in” instead of “for”, delete “which were collected”. Insert “by” before “using”. End the sentence at “…..technique” and start the other by “the height of pasture cut was 4cm”

Line 155: frozen in which temperature? Please specify

Line 157: add “feed” before “sample”

Line 167: add “twice” before “at 0700….”, add an “s” to “milk yield”

Line 168-169: what do you mean by “The average for the final week of each period is reported”? please rephrase

Line 172: add an “s” to “milking”

Results

Line 245-246: add “the” before “dietary”

Line 247:, instead of “in HMC fed cows”, write “in cows fed with HMC”

Table 4: add “composition” after “milk production”

Line 254: “however” instead of “although”

Line 256: compared to who???? Please specify

There are no line numbers after Table 5.  Please add

Discussion

Add a heading after the first paragraph in the Discussion part.

Line: Body condition score

Lines 314-316: Please clarify why is there a difference in milk fat content between studies. Also, instead of “although” write “however”

Lines 334-338: Please rephrase

Line 342: put the references at the end of the sentence

Conclusion

Please refer the recommended concentration of sugar beet silage that can be included in the ration based on your results. Also, I suggest to highlight the advantages of sugar beet silage based on your findings with regards to livestock production.

Author Response

Dear reviewer,

We would like to thank for your suggestions, which improved the quality of manuscript. The original text has modifications and we have attended your comments.

Answer for each comment was included, as described below.

regards

Dr. Rubén Guillermo Pulido

Correspondence author

Comments and Suggestions for Authors, Reviewer 

  1. The English must be improved in the whole manuscript
  2. The number of animals used for this study is too low (18), making the experimental design very weak and thus, the findings cannot strongly be supported
  3. A final paragraph in the discussion section, highlighting the main findings of this work, as well as the benefits the industry and stakeholders may obtain by these findings, is recommended. L21: change text “sugar beet silage allowed a milk production” by “sugar beet silage allowed milk production”

ANSWERS Reviewer 2

L59: incorporation in text “for” at “make herbage insufficient for”

L60: replacement in text, “restricting access time to pasture” by “Pasture restriction access time”

L63-64: replacement in text “therefore, it is necessary provide supplementary feeds to meet the nutritional requirements of the milking herd” by “Therefore, it is necessary to supplement the feeds to meet the nutritional requirements of dairy cows”

L66: replacement in text “per cow and per ha are as follows” by “per cow and per ha are the follows”

L68: replacement in text “The energy supplementation strategies include different combinations of level” by “The energy supplementation strategies include combinations of level”

L69-70: replacement in text “In addition, when the supplements were offered as a partial mixed ration (PMR)” by “In addition, when the supplements are offered as a partial mixed ration (PMR)”

L71: replacement in text “variable ruminal fluid pH” by “variable pH in ruminal fluid”

L72: replacement in text “DMI from pasture in the cows offered PMR” by “pasture DMI in cows with PMR”

L74: replacement in text at “with corn as the main energy” by “corn as the main energy”

L81-82 “Although primarily grown for sugar production, sugar beets and their by-products have been used as feed for cattle for centuries” by “Although sugar beets and their by-products were primarily grown for sugar production, they used as feeds for cattle for centuries”

L83: replacement in text “concentration of sugar” by “sugar concentrations”

L85: replacement in text “beets have been fed fresh during wintertime” by “beets have been fed during winter”

L89: replacement in text “few feeding trials to supports the feeding of sugar beets” by “few feeding trials to supports sugar beets”

L113: Table 1. Change title by “Ingredients and chemical composition of treatments”; change the formulation by “% DM” at the ingredients

L118: change text “All cows were offered an herbage allowance” by “All cows were offered a herbage allowance”

L138: replacement in text “Maize silage had a particle size of approximately 1.0 cm” by “Maize silage particles with approximately size of 1.0 cm”

L146-147: replacement in text “Pasture samples were collected three times for each study period, which were collected before cows entered the new daily strip” by “Pasture samples in triplicates in each study period, before cows entered the new daily strip”

L149: replacement in text “All samples were frozen until the laboratory analysis” by “All samples were frozen at -20°C until the laboratory analysis”

L283-284: replacement in text “The achievement of high milk yields cows depends on high DM intake and the energy concentration in the ration” by “The achievement of dairy high milk yields cows depends on high DM intake and energy concentration in the ration”

L293: replacement in text “(pure beet silages or beet silage co-ensilad with other ingredients) and due to differences in the feed” by “(pure beet silages co-ensilad with other ingredients) and to differences in feed”

L297-298: replacement in text “that the animals mobilizer body reserves to support the demands for lactogenesis” by “a mobilization of body reserves by the animals to supports the lactogenesis demands”

L305-306: replacement in text “a lower milk far concentration when molasses and barley” by “a lower concentration of milk fat when molasses and barley”

L307: replacement in text “(1994) concluded that sucrose had a lower net energy value than” by “[28] concluded that net energy contents on sucrose was lower than”

L321-322: replacement in text “Blood parameters have been reliably used to evaluate the health, nutritional, physiological, and pathological status of livestock” by “Blood parameter have been used for livestock health, nutritional, physiological, and pathological status”

L341: replacement in text “indicating an adequate nutritive balance” by “indicating a nutritive balance adequate”

L344-355-356: replacement in text “The primary concern to feeding sugar is the perception that the sugar will ferment to acids quickly, lowering rumen pH and contributing to acidosis (Evans et al., 2016)” by “With sugar feeding, the primary concern is the perception that it will ferment to acids quickly, decreasing rumen pH and contributing to acidosis [12]”

L349-350: replacement in text “In grazing dairy cows, VFA concentration in ruminal fluid can range between values of 50–150 mmol/L depending on the diet (Holmes et. al., 2002).” by “In grazing dairy cows, rumen fluid concentrations of VFA can range between 50-150 mmol/L, depending on the diet [34].”

L368-369: included text: “based on this study results”

Round 2

Reviewer 1 Report

The author has revised the article based on reviews recommendations and can be published. 

Reviewer 2 Report

The manuscript has been improved. I recommend the publication in its current form.